# Podocytopathy Associated with IgA Nephropathy in Pregnancy: A Challenging Association

**DOI:** 10.3390/jcm12051888

**Published:** 2023-02-27

**Authors:** Alejandra Orozco Guillén, Virgilia Soto Abraham, Bernardo Moguel Gonzalez, Giorgina Barbara Piccoli, Magdalena Madero

**Affiliations:** 1Nephrology Department, Instituto Nacional de Perinatología “Isidro Espinosa de los Reyes”, Mexico City 11000, Mexico; 2Department of Pathology, Ignacio Chávez National Institute of Cardiology, Mexico City 14080, Mexico; 3Department of Nephrology, Ignacio Chávez National Institute of Cardiology, Mexico City 14080, Mexico; 4Nephrologie, Department of Medicine, Centre Hospitalier Le Mans, 72037 Le Mans, France

**Keywords:** pregnancy IgA nephropathy, kidney biopsy in pregnancy, podocytopathy, immunosuppressive treatment in pregnancy

## Abstract

IgA nephropathy is the most common form of primary glomerulonephritis. While associations of IgA and other glomerular diseases have been described, the association of IgA nephropathy with “primary” podocytopathy is rare and has not been reported in pregnancy, due in part to the infrequent use of kidney biopsy during pregnancy, and a frequent overlap with preeclampsia. We report the case of a 33-year-old woman with normal kidney function, referred in the 14th gestational week of her second pregnancy, due to nephrotic proteinuria and macroscopic hematuria. The baby’s growth was normal. The patient reported episodes of macrohematuria one year previously. A kidney biopsy performed at 18 gestational weeks confirmed IgA nephropathy, associated with extensive podocyte damage. Treatment with steroids and tacrolimus led to remission of proteinuria and a healthy baby, adequate for gestational age, was delivered at 34 gestational weeks and 6 days (premature rupture of membranes). Six months after delivery, proteinuria was about 500 mg per day, with normal blood pressure and kidney function. This case highlights the importance of timely diagnosis in pregnancy and underlines that good maternal and fetal outcomes can be achieved with appropriate treatment, even in complex or severe cases.

## 1. Background

In spite of the many therapeutic advances that have been made, glomerular diseases in young individuals are always of concern, and generate uncertainty for both the patient and their family [1,2,3,4].

When a glomerulonephritis flares or is discovered during pregnancy, it represents a challenge that requires deep commitment not only from the patient and their family, but also from the medical team [5,6]. In addition, pregnancy may affect the course of glomerular diseases, eventually modulating the response to treatment [7,8,9,10,11].

Globally, IgA nephropathy is the most common form of glomerulonephritis. Several associations with other glomerular diseases have been reported [9,10,11,12]: the association with focal segmental glomerulosclerotic lesions has been extensively described, and represents an important negative prognostic element in disease progression [13]. However, the association of IgA nephropathy with “primary” podocytopathy is rare [3,14,15].

Thus far, no such case has been reported in pregnancy, possibly due to the infrequent use of renal biopsies in pregnancy and puerperium, and a frequent overlap with preeclampsia, whose differential diagnosis with glomerular diseases may be difficult or impossible to make during pregnancy without the confirmation provided by a kidney biopsy [16,17].

The case reported here highlights the importance of timely identification of different glomerular lesions in pregnancy and, in line with some previously published research results, shows that good maternal and fetal outcomes can be achieved with appropriate treatment, even in complex and severe cases [2,18,19].

## 2. The Case

Due to nephrotic proteinuria and macroscopic hematuria, in the 14th gestational week of her second pregnancy, a 33-year-old woman with normal kidney function was referred to a tertiary hospital in Mexico City.

She reported that she did not drink alcohol or use illicit drugs, but had a one-year history of cigarette smoking. Her first pregnancy, occurring 7 years previously, was uneventful: at 38 gestational weeks, she gave birth to a healthy female baby weighing 3.1 kg with normal APGAR scores (8 and 9 at 1 and 5 min, respectively).

In the previous year, her clinical history had been marked by episodes of intermittent macroscopic hematuria, edema, and hypertension, and when carefully questioned, she said that hematuria had tended to appear in concomitance with upper respiratory tract infections. She underwent a kidney biopsy, which was non-diagnostic and, despite the fact that lupus serology was negative, she received a putative diagnosis of lupus nephropathy and was treated with prednisone 50 mg for 5 weeks, followed by gradual tapering to 5 mg per day, which was maintained for 6 months. The only available control after steroid tapering was urinalysis, showing proteinuria >300 mg/dL.

Her second pregnancy was unplanned, and she had initially been advised to terminate it, prompting her to seek a second opinion in our institution.

When admitted, she was normotensive, with slight edema of the lower limbs; proteinuria was 6.6 g per day. Her urinary sediment is shown in Figure 1. The baby’s growth was normal.

While the clinical history suggested IgA nephropathy (concomitance between hematuria and upper respiratory tract infections), nephrotic proteinuria is less frequent in IgA nephropathy, and a kidney biopsy was performed at 18 gestational weeks.

While waiting for the results, she was given three pulses of 500 mg methylprednisolone, followed by oral treatment with 30 mg of prednisone per day (0.5 g/kg).

The biopsy retrieved eight glomeruli, one fully sclerotic, and four with focal and segmental lesions; the interstitium shows areas of fibrosis with associated tubular atrophy affecting 15% of the area (grade 1). Preglomerular and interstitial arteriolar vessels have moderate circumferential wall thickening, with hypertrophy of the media (Appendix A, blue arrow).

The biopsy revealed IgA nephropathy (MEST score: M1E0S1T0C0), along with focal segmental glomerulosclerosis, with podocyte hypertrophy and tip-like lesions, as shown in (Figure 2 and Figure 3). The direct immunofluorescence stain (Figure 4) shows intense IgA positivity, supporting a diagnosis of IgA nephropathy.

In Figure 5, in addition to the electrondense deposits, we can appreciate detachments of epithelial cells and podocyte foot process effacements that suggest damage to the epithelial cells. The electron microscopy image shows that there is diffuse effacement of over 80% of the podocyte processes, and there are areas of the basement membrane devoid of podocytes (podocytopenia).

In the absence of a decrease in proteinuria after 2 weeks, tacrolimus (3 mg per day) was started. Tacrolimus levels were kept between 5–6 ng/mL. The clinical response was rapid and good, allowing tapering prednisone after a further 6 weeks and, at the 30th gestational week, proteinuria decreased to 561 mg/24 h. At 34 gestational weeks and 6 days, the patient experienced a premature rupture of the membranes. A healthy female baby weighing 2484 g (Capurro 34.6 gestational weeks; APGAR 8 and 9 at 1 and 5 min, respectively) was delivered by cesarean section, performed due to a decrease in fetal heart rate during labor.

Six months after delivery, the patient’s proteinuria was stable at about 500 mg per day and her blood pressure and kidney function were normal (Table 1).

## 3. Discussion

The clinical presentation, treatment, and setting this case entailed merit reflection.

First, clinical presentation: Pregnancy is increasingly being recognized as an opportunity to diagnose glomerulonephritis and, to a lesser extent, other kidney diseases. Glomerulonephritides often have an explosive presentation, which can range from intense proteinuria to full-blown nephrotic syndrome (Table 2). This type of presentation is not exceptional in glomerular diseases, such as IgA nephropathy, that are rarely characterized by a rapid-onset nephrotic syndrome except during pregnancy [2,20,21]. In this context, it is often difficult to make a differential diagnosis based entirely on the clinical picture (Table 2). While a kidney biopsy is not without risks in pregnancy, the data on which are only available through systematic reviews that are quite old, and recent evidence, albeit in small series, suggests that in experienced hands, the procedure can be safe and have a favorable cost–benefit yield [16,17,19].

In this specific case, the differential diagnosis included diseases with different treatments, and complex and unusual pictures that may be more frequent in pregnancy than previously thought [19]. The kidney biopsy was performed in this context.

The biopsy confirmed the presence of IgA nephropathy, but also showed relevant podocyte hypertrophy, in the presence of very mild signs of glomerular sclerosis and interstitial fibrosis (Figure 1 and Figure 2).

In IgA nephropathy, glomerulosclerotic lesions, usually tip lesions, are relatively frequent and are usually a sign of secondary glomerulosclerosis caused by a critical reduction in the nephron population and the related glomerular hyperfiltration [18,20]. The clinical course of proteinuria in these cases is usually mild, and chronic lesions are usually evident in the kidney biopsy. Furthermore, a response to immunosuppressive treatment is usually absent [2].

Conversely, primary podocyte injury is characterized by a rapid and stormy clinical course with heavy proteinuria, that often responds quickly to immunosuppressive treatment.

In fact, although the histological picture suggested a primary podocyte disease associated with IgA nephropathy (podocyte hypertrophy, without relevant glomerular sclerosis and interstitial fibrosis), it did not rule out a secondary form, as sclerotic kidney lesions may be patchy and not represented in the biopsy sample. However, the prompt response to tacrolimus, which is usually ineffective in IgA nephropathy was in keeping with the presence of a “primary” form of nephrotic syndrome associated with IgA nephropathy.

The lack of regular follow-up between pregnancies, and the unavailability of the first kidney biopsy, make the interpretation of the findings even more difficult. However, the lack of nephrology follow-up, even in cases with a previous diagnosis of a kidney disease, is unfortunately frequent in particular in low-resourced settings, in which the acknowledgement of the importance of kidney diseases in pregnancy needs to be better established.

A second point is treatment choices. These are often limited and have to be balanced against risks. In this case, we considered that while macrohematuria was the hallmark of IgA nephropathy, the intense proteinuria found was probably linked to podocytopathy, and treatment was therefore combined bolus steroids with a second anti-proteinuric agent (tacrolimus). The hypothesis of a “primary” podocyropathy is indirectly supported by the prompt clinical response to therapy with tacrolimus. However, we cannot exclude that the podocytopathy was linked to IgA nephropathy, and that the unusual response was linked to pregnancy in itself. Incidentally, the choice was due to the fact that tacrolimus was readily available in our setting (unlike cyclosporine). The interpretation of blood values of calcineurin inhibitors is complex in pregnancy, due to a higher free drug dose [19,22]. In this case, the normal kidney function was reassuring, but we tried to keep tacrolimus levels as low as possible within the therapeutic window.

The reasons why pregnancy, in the context of kidney diseases, including IgA nephropathy, even in the absence of hypertensive disorders of pregnancy, is associated with preterm delivery is not known [23,24]. The association is present starting in the early CKD stages and is observed even in conditions, such as kidney donation or previous kidney stones, in which the nephron mass is or may be reduced but no active kidney disease is present [25,26].

Third, diagnostic challenges and clinical management have to take the availability of treatment in a specific setting into account. In Mexico, the health system covers the care of pregnant women, even when they are uninsured, and this leads to a more comprehensive diagnostic approach that allows us to reach a diagnosis that will also contribute to determining treatment after delivery [27]. However, in this case, due to the early severe presentation, a kidney biopsy would have probably also been performed in other contexts, without such a logistic pressure. The risk of developing superimposed preeclampsia is present in all glomerulonephritides in pregnancy, and periodic controls of the biomarkers sFlt1-PlGF make early diagnosis possible [18,20]. In spite of their acknowledged relevance, these tests are not included in the current guidelines for follow-up in high-risk pregnancies, and are not generally reimbursed. Due to cost constraints, this useful tool was not available in our setting, and our monitoring for the hypertensive disorders of pregnancy was based on the clinical picture, standard laboratory data, Doppler controls and fetal growth [2].

Monitoring podocyturia is also suggested as a tool for an early diagnosis of preeclampsia in high-risk pregnancies, but the test requires a skilled operator, is time consuming, and was not available in our case. Furthermore, in our case the results might have been affected by the underlying podocyte disorder [18].

In conclusion, this case, reporting on an unusual association between IgA nephropathy and severe, possibly primary podocytopathy discovered in pregnancy, describes the challenges involved in diagnosing and treating glomerulonephritis in pregnancy, particularly in limited-resourced settings, and points to the need for shared indications for monitoring high-risk pregnancies, which is shown to be feasible in high and low–middle income countries.

## Figures and Tables

**Figure 1 jcm-12-01888-f001:**
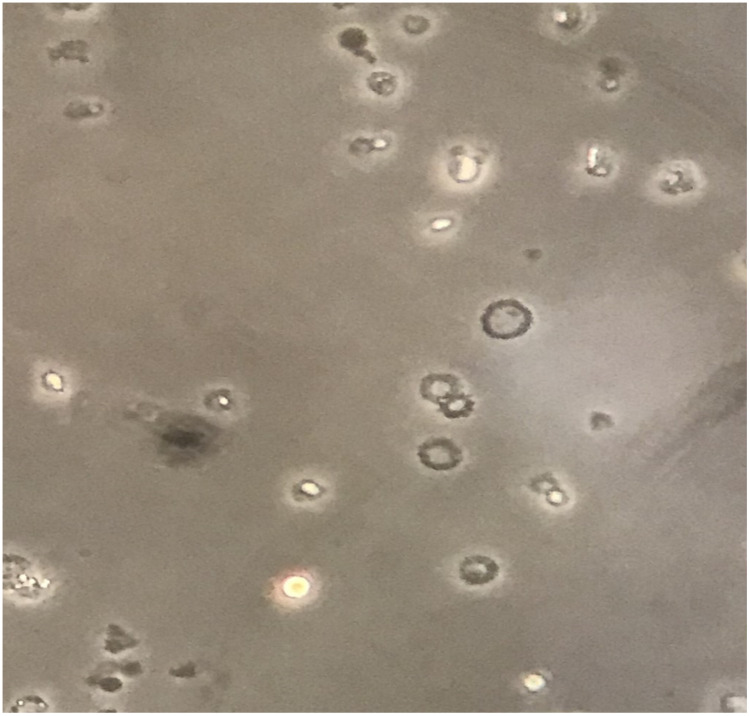
Urinary sediment, magnification 400×. Dysmorphic erythrocytes, most of which are acanthocytes, an indication of their glomerular origin.

**Figure 2 jcm-12-01888-f002:**
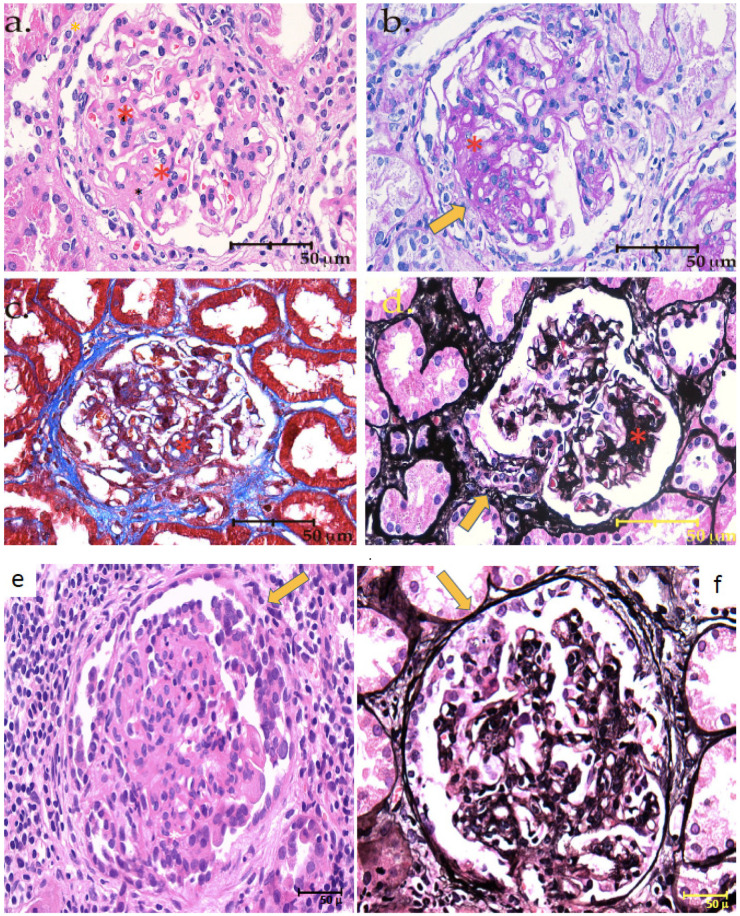
400× microphotographs stained with HE (**a**,**e**), PAS (**b**), Masson’s trichrome (**c**), and Jones’ methenamine silver (**d**,**f**), respectively, in which different glomeruli with global mesangial widening and proliferation (M1) with cicatricial segmental sclerosing lesions (S1) are observed). Asterisks show mesangial proliferation (**a**–**d**). Arrows shows segmentary sclerosis in the tip (**b**,**d**). In the last two images we can see glomeruli with podocyte hypertrophy, forming crowns around some segments of the capillaries (arrows).

**Figure 3 jcm-12-01888-f003:**
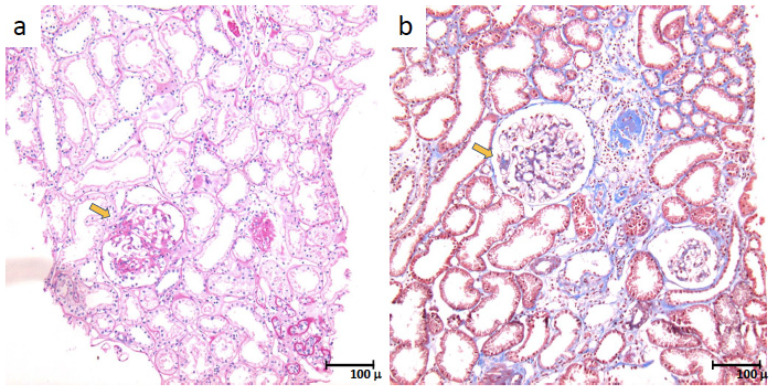
Photomicrographs at 100× stained with PAS (**a**) and Masson’s trichrome (**b**), respectively, in which the segmental lesions are indicated at low magnification (yellow arrows).

**Figure 4 jcm-12-01888-f004:**
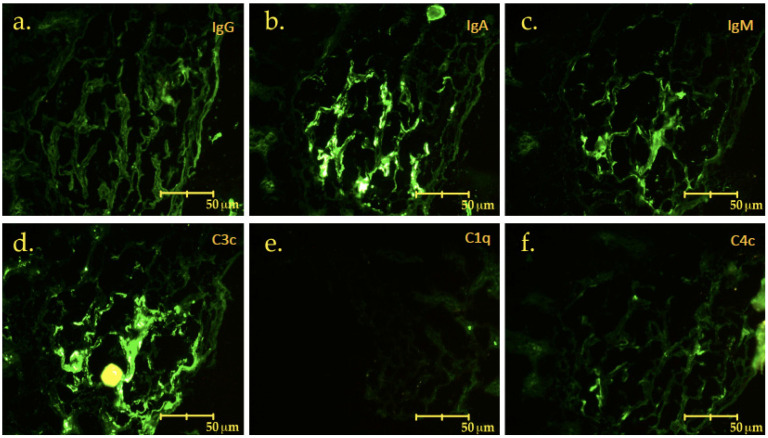
400× direct immunofluorescence study that showed dominance positivity for IgA (**b**), with little IgM (**c**) and mesangial deposits of C3 (**d**) that allow us to conclude with the diagnosis of IgA nephropathy. (**a**) IgG, (**e**) C1q, (**f**) C4c.

**Figure 5 jcm-12-01888-f005:**
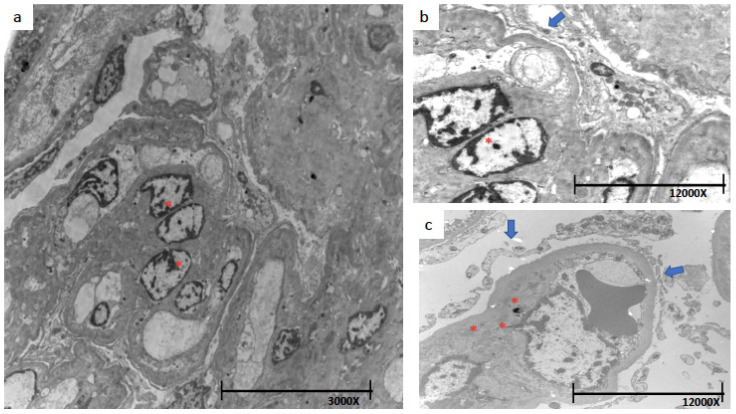
(**a**) 3000× Electron microscopy shows mesangial hypercellularity *, (**b**,**c**) 12,000× electron microscopy. Shown in b, mesangial hypercellularity * and effacement of podocyte processes (arrow), (**c**) There is accentuated damage to podocyte processes (arrow) with deposits of electron dense material (immune complexes) especially in the mesangium (*).

**Table 1 jcm-12-01888-t001:** Biochemical data from referral to delivery and after pregnancy.

	At Referral	At Kidney Biopsy	30 Gestational Weeks	Delivery	6 Months after Delivery
Blood pressure	110/70	120/80	120/80	130/80	110/60
Haematuria	++++	++++	++	++	++
SCr (mg/dL)	0.8	1.0	0.7	0.8	1.0
BUN (mg/dL)	17.8	24.6	15	19.6	28
e-GFR (ml/min)	100	76	117	100	76
Albumin (g/dL)	2.5	1.9	2.9	3.5	4
Total proteins	5	4.2	5.3	6.7	7
Proteinuria (mg/24 h)	6600	7524	562	736	500
Albuminuria (mg/24 h)	3121	4232	561	na	112
Hemoglobin (g/dL)	12.4	12.6	12.5	11.6	13
Platelets (mm^3^)	273	232	363	327	309
Tacrolimus ng/mL	-	-	4	5.6	6
Immunologic markers: C3 66 mg/dL (normal values: 66–185 mg/dL); C4 16 mg/dL (normal values: 12–49 mg/dL), ANA IFI > 1:40 B2 glycoprotein (IgG) 2.9 CU, B2 glycoprotein (IgM) 1.7 CU, anticardiolipin (IgM) 1.4 CU, DNA double strain antibodies 16.17 UI/mL (<35 UI/mL), Antibodies Anti La 2.72, Antibodies anti Ro 2.67					
Viral markers: Hepatitis B and C, and HIV negative					

**Table 2 jcm-12-01888-t002:** Differential diagnosis of proteinuria and hematuria in early pregnancy in our patient.

Sign/Symptom	Diagnosis	Suggests	Excludes	Likelihood
Proteinuria at 14 gestational weeks	Preeclampsia	Onset may be anticipated in multiple pregnancies; in rare cases proteinuria antedates hypertension and full-blown preeclampsia	Preeclampsia is defined by an onset after 20 gestational weeks. The absence of hypertension excludes this diagnosis.	−−−
	Pregnancy-induced proteinuria	The diagnosis is based on the association with pregnancy and subsequent regression.	Generic term. Cannot be diagnosed during pregnancy, only afterwards.	−+−
	Glomerulonephritis	Early onset proteinuria, history of macrohematuria.	No exclusion.	+++
Hematuria (macroscopic, possible association with infections in the past), associated with proteinuria	IgA nephropathy	History of macrohematuria concomitant with infections. Frequent association with proteinuria in pregnancy.	Possible, but the nephrotic syndrome is early and unusually severe; high grade proteinuria is often seen in the last trimester.	+−+
	Minimal change-focal segmental glomerulosclerosis	Rapid onset, high grade proteinuria with nephrotic syndrome. Compatible with microhematuria, in particular in pregnancy.	Macrohematuria is very rare in this context.	+−+
	Membranous and proliferative glomerulonephritis	Typically combines hematuria and proteinuria.	Rare in the absence of hypertension and with normal kidney function.	+−−
	Lupus nephropathy	May combine hematuria and proteinuria; common in women of childbearing age.	Very rare in the absence of positive lupus serology (repeated over time).	+−−
	Other complex pictures	May be relatively frequent in pregnancy, possibly underestimated. May combine different pictures; pregnancy may modulate presentation, usually with a high proteinuria level.	No exclusion.	+−+

## Data Availability

Not applicable.

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
