# Peer review of "Podocytopathy Associated with IgA Nephropathy in Pregnancy: A Challenging Association"

_jcm, 2023, doi:10.3390/jcm12051888_

Round 1
Reviewer 1 Report (Previous Reviewer 1)
The authors attempted to revise the manuscript appropriately in response to my suggestions. I feel the sincerity. However, there are some serious problems in this manuscript after revision.
(1) The authors indicated tip lesions by arrows in Figure 2b and 2d. However, I can not agree with them. Therefore, the authors should show the convincing pictures.
(2) The authors indicated podocyte hypertrophy by arrows in Figure 2e and 2f. However, I can not agree with it. Therefore, the authors should show the convincing pictures.
(3) The authors indicated foot process effacement by arrows in Figure 4b and 4c. However, I can not agree with it. Therefore, the authors should show the convincing pictures.
Author Response
Thanks you for taking the time to revise again our manuscript.
We tried to answer by supplying different images, and adding one supplemental EM image.
We think that the arrows were sometimes displaced in the previous images, and we apologize for this.
However, would the reviewer have further comments, we would kindly ask to be more specific. In fact a statement as “I can not agree” or “the authors should show the convincing pictures” is of no specific use for improving or changing, and each time we have to review all our images, we have to reserve a pathology session, which is not immediate and is the reason why we are submitting only now the revised paper.

Reviewer 2 Report (Previous Reviewer 2)
I think the revised paper is very readable and well received. I am happy to review this paper.
However, I would like to make a few changes.
Primary podocyte injury does not always respond to immunosuppressive drugs and may present with refractory nephrotic syndrome. Therefore, please change "often" to "usually" on page 10, line 169.
The differential disease in this case may be focal segmental glomerulosclerosis secondary to pregnancy. It usually does not improve without delivery or interruption of pregnancy. In the present case, the response to treatment while continuing the pregnancy is an important clinical indication in favor of primary podocyte injury. Please add to the discussion.
Please correct in three places.
Page 3, line 78.
Please change "interstitum" to "interstitium" .
Page 10, line 189.
Please change "directionally" to "directly".
Page 10, line 192
Please change "reposnse" to "response".
Author Response
Thank you for the advice, we did it.
The differential disease in this case may be focal segmental glomerulosclerosis secondary to pregnancy. It usually does not improve without delivery or interruption of pregnancy. In the present case, the response to treatment while continuing the pregnancy is an important clinical indication in favor of primary podocyte injury. Please add to the discussion

Reviewer 3 Report (Previous Reviewer 3)
The Authors have modified the manuscript, answering all the questions and suggestions posed.
Author Response
The Authors have modified the manuscript, answering all the questions and suggestions posed.
Round 2
Reviewer 1 Report (Previous Reviewer 1)
The authors attempted to revise the manuscript appropriately in response to my suggestions. I appreciate the authors’ efforts. I have no special concerns about this revised manuscript.
This manuscript is a resubmission of an earlier submission. The following is a list of the peer review reports and author responses from that submission.
Round 1
Reviewer 1 Report
The authors indicated the association of IgA nephropathy with primary podocytopathy. This manuscript is important, because kidney biopsy is infrequently performed during pregnancy. In addition, because the original renal damage is influenced to no small extent by preeclampsia, a more active engagement is needed to elucidate the pathology in the pregnant patients. However, there are some concerns in this manuscript.
(1) The authors concluded that podocytopathy is primary but not secondary, because tip lesion is observed despite the absence of marked sclerosing lesions and tubulointerstitial damages and a response to immunosuppressive treatment with tacrolimus is usually poor. However, it is reported that tacrolimus is effective to IgA nephropathy. In addition, segmental sclerotic lesions may simply not be seen on the specimen. Therefore, I think that it is difficult to conclude that podocytopathy is primary. However, if the authors would like to emphasize it, it is necessary to describe in detail how many glomeruli were collected, how many global sclerosis and segmental sclerosis were present, and what percentage of tubulointerstitial damages were present. In addition, glomeruli with tip lesions should be shown in a figure.
(2) The authors indicated the objective lens magnification in this case report. However, the magnification of the microscope should be expressed as the overall magnification, not the objective lens magnification.
(3) The authors only indicated the presence of electron dense deposits in Figure 4. The authors should show whether there are detachments of epithelial cells from glomerular basement membrane or foot process effacements suggestive of epithelial cell damage.
(4) Although the authors indicated that blood pressure was normal, actual blood pressure levels are necessary. Therefore, the authors should add the information in Table 2.
(5) The authors did not show the condition of hematuria. Therefore, the authors should add the information in Table 2.
(6) I have no idea what kind of staining it is, because it is not indicated. Authors should add stain names in Figure 2 legends. In addition, abnormal findings should be indicated in Figure 2 using asterisks or arrows.
(7) It is unclear which pictures were stained with which antibodies. Therefore, the authors should add the information in Figure 3.
(8) The authors should present pictures of PAS staining rather than HE staining to reveal podocyte abnormalities in Figure 5. In addition, photographs of serial sections are good for PAS staining and PAM staining in Figure 5.
(9) The authors should explain why proteinuria decreased dramatically after the use of tacrolimus but then leveled off.
Author Response
The authors indicated the association of IgA nephropathy with primary podocytopathy. This manuscript is important, because kidney biopsy is infrequently performed during pregnancy. In addition, because the original renal damage is influenced to no small extent by preeclampsia, a more active engagement is needed to elucidate the pathology in the pregnant patients. However, there are some concerns in this manuscript.
- The authors concluded that podocytopathy is primary but not secondary, because tip lesion is observed despite the absence of marked sclerosing lesions and tubulointerstitial damages and a response to immunosuppressive treatment with tacrolimus is usually poor. However, it is reported that tacrolimus is effective to IgA nephropathy. In addition, segmental sclerotic lesions may simply not be seen on the specimen. Therefore, I think that it is difficult to conclude that podocytopathy is primary. However, if the authors would like to emphasize it, it is necessary to describe in detail how many glomeruli were collected, how many global sclerosis and segmental sclerosis were present, and what percentage of tubulointerstitial damages were present. In addition, glomeruli with tip lesions should be shown in a figure.
Answer: Thank you for your comments. The biopsy included 8 glomeruli, of which 1 had global sclerosis and tubulointerstial damage was present in 15% of the sample.
We agree that there is no conclusive way to demonstrate that these lesions are independent of IgA nephropathy. We smoothed the description, limiting it to the lesions and the clinical response. Hypertrophic and vacuolated podocytes were present. Four glomeruli presented segmental lessions (3 located at the tip).
Furthermore, we smoothed the diagnostic comments, as follows:
In the introduction:
The case reported here highlights the importance of timely identification of different glomerular lesions in pregnancy and, in line with some previously published research results, shows that good maternal and fetal outcomes can be achieved with appropriate treatment even in complex and severe cases (18-20).
In the discussion:
The biopsy confirmed the presence of IgA nephropathy, but also showed relevant podocyte hypertrophy, in the presence of very mild signs of glomerular sclerosis and interstitial fibrosis (Figures 1, 2).
And
In fact, although the histological picture suggested a primary podocyte disease associated with IgA nephropathy (podocyte hypertrophy, without relevant glomerular sclerosis and interstitial fibrosis), it did not rule out a secondary form, as sclerotic kidney lesions may be patchy and not represented in the biopsy sample. However, the prompt response to tacrolimus, which is usually ineffective in IgA nephropathy was in keeping with the presence of a “primary” form of nephrotic syndrome associated with IgA nephropathy.
(2) The authors indicated the objective lens magnification in this case report. However, the magnification of the microscope should be expressed as the overall magnification, not the objective lens magnification.
Thank you for this comment. The correct magnification is 100X and 400X. It was corrected in the legends.
(3) The authors only indicated the presence of electron dense deposits in Figure 4. The authors should show whether there are detachments of epithelial cells from glomerular basement membrane or foot process effacements suggestive of epithelial cell damage.
Thank you for your comment.
We added it in the text the following description:
In figure 4, beside electrondense deposits, we can appreciate detachments of epithelial cells and podocyte foot process effacements that suggest damage to the epithelial cells. The electron microscopy image shows that there is diffuse effacement of over 80% of the podocyte processes and there areas of basement membrane devoid of podocytes (podocytopenia).
- Although the authors indicated that blood pressure was normal, actual blood pressure levels are necessary. Therefore, the authors should add the information in Table 2.
Thank you for this comment, we added blood pressure in Table 2.
- The authors did not show the condition of hematuria. Therefore, the authors should add the information in Table 2.
We added the information on hematuria to table 2; thank you for pointing this out.
- I have no idea what kind of staining it is, because it is not indicated. Authors should add stain names in Figure 2 legends.
Thank you for the comment, we add the type of stain used in each photo.
- In addition, abnormal findings should be indicated in Figure 2 using asterisks or arrows.
We added arrows and asterisks to highlight the findings. Thank you for the suggestion.
- It is unclear which pictures were stained with which antibodies. Therefore, the authors should add the information in Figure 3.
Thanks for this comment, we added the information
- The authors should present pictures of PAS staining rather than HE staining to reveal podocyte abnormalities in Figure 5. In addition, photographs of serial sections are good for PAS staining and PAM staining in Figure 5.
Unfortunately for this case we do not have these serial cuts and the block does not have more material because the tissue was taken for the electronic microscopy.
The authors should explain why proteinuria decreased dramatically after the use of tacrolimus but then leveled off.
As you correctly underlined we can only make hypotheses. Our is that this occurred because the proteinuria linked to podocytopathy responded to treatment while the component linked to IgA nephropathy did not.
We added the following remarks in the text
In this case, we considered that while macrohematuria was the hallmark of IgA nephropathy, the intense proteinuria found was probably linked to podocytopathy, and treatment therefore combined bolus steroids with a second anti-proteinuric agent (tacrolimus). The hypothesis of a “primary” podocyropathy is indirectely supported by the prompt clinical response to therapy with tacrolimus. However, we cannot exclude that the podocytopathy was linked to IgA nephropathy, and that the unusual reposnse was linked to pregnancy in itself.
Thank you for the comments and suggestions to improve the quality of our study.
Reviewer 2 Report
This case report is unusual in that a renal biopsy was performed in a patient who developed nephrotic syndrome during pregnancy.
It is also very impressive that the patient was able to achieve remission during pregnancy by combining steroid therapy and immunosuppressive therapy during pregnancy.
However, it is not uncommon for nephrotic syndrome to be complicated by secondary focal segmental glomerulosclerosis when the patient presents with nephrotic syndrome during pregnancy. When a renal biopsy is performed because of nephrotic syndrome, IgA nephropathy is sometimes associated with minimal change disease or focal segmental glomerulosclerosis.
Although it is interesting that this case occurred during pregnancy, I have some doubts.
I have difficulty in recognizing the segmental sclerosis and podcytepathy in the renal biopsy histopathology images in Figures 2 and 5.
Tip lesions are usually found at the tubular pole, but do not appear to be at the tubular pole.
The findings of the histopathology of the renal biopsy are insufficient.
Number of all glomeruli sampled
Number of glomeruli with global sclerosis
Number of glomeruli with segmental sclerosis
Evaluation of interstitium
Evaluation of blood vessels
Histopathology images taken at smaller magnifications should also be included.
In Figure 3, it is not clear which antibody was used to stain each picture.
I do not understand why Figure 2 and 5 are separated.
Is the magnification in Figure 2 and 5 40x, or is it 400x?
We do not know the results of the first renal biopsy, but is it safe to assume that the patient had severe proteinuria at least before the second pregnancy? Based on the clinical course, do you think this case is a relapse of FSGS?
If the patient had a high level of proteinuria before pregnancy, and if it was aggravated by pregnancy, this is not an uncommon case.
This case report is very interesting, but the clinical course has not yet been fully summarized and discussed.
Author Response
Comments and Suggestions for Authors
This case report is unusual in that a renal biopsy was performed in a patient who developed nephrotic syndrome during pregnancy.
It is also very impressive that the patient was able to achieve remission during pregnancy by combining steroid therapy and immunosuppressive therapy during pregnancy.
However, it is not uncommon for nephrotic syndrome to be complicated by secondary focal segmental glomerulosclerosis when the patient presents with nephrotic syndrome during pregnancy. When a renal biopsy is performed because of nephrotic syndrome, IgA nephropathy is sometimes associated with minimal change disease or focal segmental glomerulosclerosis.
Although it is interesting that this case occurred during pregnancy, I have some doubts.
I have difficulty in recognizing the segmental sclerosis and podcytepathy in the renal biopsy histopathology images in Figure 2.
Thank you for your comment. We have added arrows and asterisks to highlight sclerotic lesions and mesangial proliferation in figure 2.
Tip lesions are usually found at the tubular pole, but do not appear to be at the tubular pole.
Thanks for this observation. Tip lesions can be seen in Jones methenamine silver Figure 2 photograph (b) and (d) . As you suggested, we added arrows.
The findings of the histopathology of the renal biopsy are insufficient.
Thank you for this observation, we’ll answer all the question below
Number of all glomeruli sampled There were 8 glomeruli.
Number of glomeruli with global sclerosis One glomerulus with global sclerosis
Number of glomeruli with segmental sclerosis we found 4 glomeruli with focal and segmental lesions;
Evaluation of interstitium: The interstitum shows bands and areas of fibrosis with associated tubular atrophy affecting 15% of the area (grade 1)
Evaluation of blood vessels: Preglomerular and interstitial arteriolar vessels have moderate circumferential hyaline thickening of the wall, with hypertrophy of the media as we can see in this image in the arrow in blue.
The following was added in the case description:
The biopsy retrieved 8 glomeruli, 1 fully sclerotic and 4 with focal and segmental lesions; the interstitum shows areas of fibrosis with associated tubular atrophy affecting 15% of the area (grade 1). Preglomerular and interstitial arteriolar vessels have moderate circumferential wall thickening, with hypertrophy of the media (supplemental figure 1, blue arrow).
Suplemental figure 1 was also added:
Supplemental figure 1: (HE staining, 400x)Preglomerular and interstitial arteriolar vessels have moderate circumferential wall thickening, with hypertrophy of the media (supplemental figure 1, blue arrow).
Histopathology images taken at smaller magnifications should also be included. Thank you for this observation. We added two images in figure 5 with smaller magnifications.
In Figure 3, it is not clear which antibody was used to stain each picture. Thank you very much for this observation, we added which antibody was used in each photo.
I do not understand why Figure 2 and 5 are separated. Thanks for this observation we combined them and added a new figure at low magnification.
Is the magnification in Figure 2 and 5 40x, or is it 400x? Thanks for commenting is 400x
We do not know the results of the first renal biopsy, but is it safe to assume that the patient had severe proteinuria at least before the second pregnancy? Based on the clinical course, do you think this case is a relapse of FSGS?
Thank you for this comment, we believe that it is possible, although we cannot be sure about it, because she she was not regularly and correctly followed up before pregnacy. We added a short comment on thisfrequent ptoblem in patients referred with kidney problems in pregnancy.
The lack of regular follow-up between pregnancies, and the unavailability of the first kidney biopsy make the interpretation of the findings even more difficult. However, the lack of nephrology follow-up, even in cases with a previous diagnosis of a kidney disease, is unfortunately frequent, in particular in low-resourced settings, in which the acknowledgement of the importance of kidmey diseases in pregnancy needs to be better established.
If the patient had a high level of proteinuria before pregnancy, and if it was aggravated by pregnancy, this is not an uncommon case.
As previously commented, we think that the podocytopathy appeared or worsened in pregnancy, although we cannot be sure.
This case report is very interesting, but the clinical course has not yet been fully summarized and discussed.
Thank you very much for your help inmproving our case report.
We appreciate the requests and hope to have fulfilled them.
Reviewer 3 Report
The Authors describe a case of IGAN associated with podocytopathy in pregnancy. This association is well described and it corresponds to what the kidney biopsy showed: the classical lesion of podocytes' hypertrophy, tip lesions, and focal segmental glomerulosclerosis (S1). It is supposed that this kind of lesion may respond to immunosuppressant therapy. Why do the Authors hypothesize a "primary" podocytopathy, without any other element? In the discussion, they offer a different version: the sclerotic lesions were not relevant, and thus they speculate a "primary" podocytopathy, forgetting that, in pregnancy, the hyperfiltration may multiply proteinuria. Moreover, the classification of podocytopathies in primary and secondary is problematic and a pathogenetic classification is suggested (Ahn w and Bomback A, AJKD, 75, P955, 2020).
I think that the reported case is important because it underlines the importance of kidney biopsy also in pregnancy, but a more coherent interpretation is necessary.
Author Response
Thank you for your comment.
We agree that there is no conclusive way to demonstrate that these lesions are independent of IgA nephropathy. We smoothed the description, limiting it to the lesions and the clinical response.
Furthermore, we smoothed the diagnostic comments, as follows:
In the introduction:
The case reported here highlights the importance of timely identification of different glomerular lesions in pregnancy and, in line with some previously published research results, shows that good maternal and fetal outcomes can be achieved with appropriate treatment even in complex and severe cases (18-20).
In the discussion:
The biopsy confirmed the presence of IgA nephropathy, but also showed relevant podocyte hypertrophy, in the presence of very mild signs of glomerular sclerosis and interstitial fibrosis (Figures 1, 2).
And
In fact, although the histological picture suggested a primary podocyte disease associated with IgA nephropathy (podocyte hypertrophy, without relevant glomerular sclerosis and interstitial fibrosis), it did not rule out a secondary form, as sclerotic kidney lesions may be patchy and not represented in the biopsy sample. However, the prompt response to tacrolimus, which is usually ineffective in IgA nephropathy was in keeping with the presence of a “primary” form of nephrotic syndrome associated with IgA nephropathy.